# A Literature Review and Gap Analysis of Emerging Technologies and New Trends in Gambling

**DOI:** 10.3390/ijerph17030744

**Published:** 2020-01-23

**Authors:** Sharon Lawn, Candice Oster, Ben Riley, David Smith, Michael Baigent, Mubarak Rahamathulla

**Affiliations:** College of Medicine and Public Health, Flinders University, Adelaide 5042, SA, Australia; candice.oster@flinders.edu.au (C.O.); ben.riley@flinders.edu.au (B.R.); david.smith@flinders.edu.au (D.S.); michael.baigent@flinders.edu.au (M.B.); mubarak@flinders.edu.au (M.R.)

**Keywords:** internet gambling, simulated gambling, sports betting, advertising, electronic gaming machines

## Abstract

There have been significant changes in the gambling landscape particularly relating to gambling in the digital age. As the gambling landscape changes, regulation of gambling also needs to change. In 2018, the Office of Responsible Gambling in New South Wales, Australia, commissioned a gap analysis to inform their research objectives and priority focus areas. This included an identification of gaps in our understanding of emerging technologies and new trends in gambling. A gap analysis of the peer-reviewed literature published since 2015 was undertaken, identifying 116 articles. The main area of focus was Internet gambling, followed by articles exploring the relationship between video gaming and gambling, the expansion of the sports betting market, Electronic Gambling Machines characteristics and articles exploring new technologies and trends in advertising and inducements. Key gaps related to the need for more research in general, as well as research focusing on subpopulations such as those using different gambling formats, those with varying levels of problem gambling, and vulnerable populations. From a methods perspective, researchers saw the need for longitudinal studies, more qualitative research and improved outcome measures. The development and testing of a public health approach to addressing the harms associated with gambling in these areas is needed.

## 1. Introduction

Technological revolutions in recent years have focused primarily on inventing ways to increase the visual, auditory and tactile stimulation of human beings. Through creative use of sound, touch, screens and visual effects the digital world has innovated many ways of captivating and controlling human emotions [1]. The gambling industry is one of the pioneers to embrace this technology. By creatively navigating audio-visual technology such as surround sound, touch screens, haptic actuators and augmented reality, among others, the gambling industry has designed gambling experiences to stimulate the human senses [2].

Using digital tools such as the Internet, Internet games and mobile phone technology, the gambling industry has expanded its customer base because of the easy access these tools can give to gambling platforms [3]. Major social media and online gaming companies have started making inroads into gambling business. This ‘digital convergence’ has created opportunities for the gambling industry to expand its customer base, particularly among young people [4]. Ease and cost effectiveness in accessing the Internet has prompted the gambling industry to invest heavily in emerging technological tools. For example, the prediction of total market value of global mobile phone gambling industry was $23 billion in 2011 compared to $2 billion in 2006 [5]. This prolific growth in online gambling has been accompanied by growing concern for its potential harms [6]; hence reviewing the current literature and evidence gaps is important and timely. Growth of online gambling has also been implicated in the general growth in gambling, more broadly; for example, in the growth of sports betting among people who did not previously gamble [7]. With such growth and potential greater diversity in gambler populations, there is likely growth in new population groups experiencing problem gambling. This rapidly evolving area of gambling has, therefore, also brought new challenges globally for regulators and policy makers, in addition to communities and problem gambling treatment providers, requiring potentially new approaches to understanding and addressing gambling harm arising from these new technologies associated with gambling.

There has been international support for a broader public health approach to address the harms associated with gambling [8,9]. As the gambling landscape changes, regulation of gambling also needs to change. In New South Wales (NSW), Australia, the Office of Responsible Gambling (ORG) develops and implements programs and initiatives as part of a strategic approach that supports responsible gambling and prevents and minimises the risk of gambling related harm in the community. The Office supports the Responsible Gambling Fund, which plays a key role in advising the NSW Government on the allocation of funds for initiatives and programs that support responsible gambling and help reduce gambling related harm. In 2018, the ORG commissioned the authors to undertake a gap analysis to inform their Research Agenda, which articulates research objectives and priority focus areas for the period 2019–2021.

Within that larger review, the ORG identified the following question as one area of interest: ‘What are the gaps in our understanding of emerging technologies and new trends in gambling?’ This understanding is crucial for evidence-based public health policymaking internationally on both promotion of responsible gambling and prevention of problem gambling. The ORG identified 2015 to 2018 as the publication timeframe in order to ensure the currency of the gap analysis.

## 2. Materials and Methods

As Robinson, Salhanha and Mckoy [10] (p. 1325) point out, the “clear and explicit identification of research gaps is a necessary step in developing a research agenda, including decisions about funding and the design of informative studies”. Unlike systematic reviews, there is no single accepted approach for conducting a gap analysis, with multiple and varied approaches used [8]. In this study we used the following approach, modified from Otto et al. [11]:Systematic literature search of peer-reviewed literature published since 2015;Summary of existing areas of research focus identified in the literature;Identification of gaps noted within existing research (as identified by the authors);Identification of gaps across the body of research;Consultation between subject matter experts on our research team and the ORG to identify further gaps not identified in the body of research;Consultation between the research team and the ORG to finalise and prioritise gaps.

The systematic literature search followed a rapid review methodology, a variant of a systematic literature review that includes a systematic search of the literature, but imposes limitations to its breadth (e.g., in terms of time, language and a reference list search) and it does not involve a meta-analysis. While rapid reviews are undertaken more quickly than systematic reviews, they are reported to produce similar conclusions to systematic reviews [12].

Peer reviewed literature was sourced through searches of the electronic databases Medline, Emcare, PsycINFO, SCOPUS, Web of Science and Proquest (Health and Medicine, Social Sciences Collection). The search terms used were identified in consultation with a research librarian. Search terms relating to emerging technologies and new trends included ‘Gambling’, ‘Betting’, ‘Wagering’, ‘Pokie’, ‘Lottery’, ‘Casino’, ‘Keno’, ‘Machine’, ‘Video Games’, ‘Technology or Information Technology’, ‘Trend/emerging trend/future/interactive/innovation in Gambling Technology’ and ‘Computer games’. The database search was conducted on 16 October 2018. The literature review was conducted between October 2018 and March 2019.

Inclusion and exclusion criteria are summarised in Table 1. Notably, unlike a systematic review where the focus is on research articles, as a gap analysis, we included commentary articles to inform future areas of research.

After removing duplicates, three reviewers conducted the initial screening for inclusion based on the information in the title and abstract. These reviewers were the first three authors; they included two senior researchers with previous experience in conducting several literature reviews, and a senior researcher with specific expertise in gambling treatment and research with some prior experience in conducting literature reviews. To ensure fidelity of the screening process, the reviewers independently reviewed title and abstract for a random sample of 50 of the citations. They then met to exchange their individual decisions and discussed their rationale for those decisions. Consensus was determined as being where two or more of the three reviewers agreed on the citation’s inclusion or exclusion. Full text articles for each included citation were then collected, with the results divided into three and screened by the three reviewers against the inclusion/exclusion criteria. As this was a rapid review, where a full text of an article was not readily available, we did not contact the author. Such studies were excluded. The reviewers met to discuss any articles where the reviewer was unsure. Information extracted from the articles included: Author names, year, and study location; Aim; Key findings; Gaps identified by the authors. Gaps were furthermore differentiated by type using the typology presented in Table 2, which was formulated by the researchers as part of the larger study, and drawn from gap types evident from across the articles. Quality ratings were undertaken for all included peer-reviewed articles. The quality of the studies was assessed using the Mixed Methods Appraisal Tool (MMAT) for qualitative, quantitative randomised controlled trials, quantitative non-randomised trials, quantitative descriptive studies, and mixed methods studies [13]. We determined that all peer-reviewed research was generally well-conducted and met the rating criteria. No studies were excluded due to poor quality (see Appendix A for detail).

## 3. Results

Search results are presented in Figure 1.

In all, 116 peer-reviewed articles published since 2015 were identified. The majority of the articles (*n* = 44) addressed Internet gambling, followed by those exploring the relationship between gaming and gambling (*n* = 24), the expansion of the sports betting market (*n* = 22), various characteristics of Electronic Gambling Machines (EGMs) (*n* = 15), and those exploring new technologies and trends in advertising and inducements (*n* = 11). The articles were predominantly cross-sectional in design (*n* = 58; 50%) and just over a third of the articles (39%) were from Australia. The next largest contribution came from Canada (14%), then USA (12%), with the remaining 18 countries contributing less than 5% each (see Table 3).

The gap analysis process led to the identification of five priority areas: Internet gambling; video gaming and gambling; electronic gaming machines; advertising and inducements; and expansion of the sports betting market. Articles relating to each area are summarised as Appendix A.

### 3.1. Internet Gambling

The area of Internet (or online) gambling is a key technological trend in gambling and was the focus of 46 peer-reviewed articles. The articles focused predominantly on the characteristics of Internet gamblers and their gambling behaviour in the online environment, with one literature review exploring the use of behavioural tracking and big data studies of Internet gambling [14]. The studies have various foci and findings, but overall demonstrate that Internet gamblers are not a homogeneous group [15].

The rise of Internet gambling has seen increasing concern that Internet gambling is more harmful than land-based gambling. This was supported in three studies [16,17,18]. However, other studies suggest that Internet gambling is not inherently more harmful than land-based gambling; issues relate more to the diversity of gambling formats [19,20], subgroups of gamblers [21,22] and modes of access [23]. Looking at younger gamblers, a Canadian study found that adolescents who gambled online were reported to be five times more at risk of problem gambling than those who gambled on land-based modes only [24]. Similar rates were found in an Italian study [25]. On the other hand, a study conducted in Hong Kong reported that adolescent Internet gamblers were at no greater risk than adolescent land-based gamblers [26].

For example, in Blaszczynski et al.’s [20] Australian survey of 4594 respondents, mixed gamblers (those that gambled online and on land) had higher problem gambling scores, while land-based gamblers experienced higher psychological distress. They concluded: “exclusive online gamblers represent a different subpopulation at lower risk of harm compared to gamblers engaging in multiple forms” [20] (p. 261). The association between problem gambling and other types of gambling was the focus of the research by Gainsbury, Russell, Blaszczynski and Hing [27] and these authors reported higher proportions of problem gambling in land-based gamblers when compared with Internet-only and mixed-mode gamblers. Gainsbury, Liu, Russell and Teichert [23] found an association between mode of access and gambling problems, with lower rates of problems among Internet gamblers who prefer to gamble using computers than those using mobile and supplementary devices.

The literature furthermore suggests a number of differences between those who gamble online to land-based gamblers. These include behavioural differences such as greater frequency of gambling in Internet gamblers [28,29,30], playing more hands and committing more errors [31] and greater debt [32]. Wijesingha et al. [33] found adolescents with low to moderate or high problem gambling severity were significantly more likely to gamble online.

Demographic background of people engaged in online versus land-based gambling was also a major theme of the research studies. For example, in a study comparing e-sports versus sports bettors, Gainsbury, Abarbanel and Blaszczynski [29] found more females and those from Asian ethnic backgrounds involved in e-sports compared to white males in sports betting. Studies have also found differences in risk factors across Internet and land-based gamblers [34], such as easy access and intangibility of money for Internet gambling versus peer group and alcohol availability for land-based gambling [35,36].

Further exploration of the association between Internet gambling and problem gambling, this time focusing only on Internet gamblers, suggests gambling via the Internet is not inherently problematic but rather appears to affect different gamblers in different ways. For example, in a recent study surveying 3199 online gamblers about their engagement with and perceptions of offshore online gambling sites, most preferred domestic sites, with offshore gamblers being more involved gamblers with greater problem gambling severity [37]. A survey of 3178 Internet gamblers demonstrated an association between having multiple accounts and gambling on more activities and more frequently, being more involved and having higher rates of gambling problems, suggesting two types of Internet gamblers (volatile vs. stable gamblers) [38]. Another survey of 2799 Internet gamblers compared problem with non-problem and at-risk Internet gamblers, identifying differences in demographics, gambling behaviour and gambling beliefs. A key finding was that Internet problem gamblers also had problems relating to land-based gambling [22]. Taking a different approach, Hing et al. [39] explored the responses of Internet sports bettors to gambling promotions during sports broadcasts. Survey results suggest a link between problem gambling in young male Internet sports bettors and positive attitudes to gambling sponsors. Again, this research points to Internet gamblers being a non-homogenous group.

A further area of focus in the literature on Internet gambling is mode of access. Gainsbury et al. [38] investigated gambling prevalence and the relationship between various gambling activities and interactive modes of access, finding 9.4% of Internet gamblers preferred using a mobile/smart phone but the majority (87.1%) preferred using a computer or laptop. A further two studies report the potential for greater problems when gambling via mobile/smart phones [40,41], reflecting the results of Gainsbury et al. [23], discussed above. Another study [42] examining the screen interface suggested immersion variables made it possible to understand the cognitive participation of individuals towards screens, in general.

#### Gap Analysis

Results of the gap analysis for Internet gambling are presented in Table 4. Despite the large body of existing research in the area of Internet gambling, knowledge gaps were the most frequently identified gap type, in particular the need for research on psychological factors, gambling intentions, behaviour or actions, and gambling outcomes. Authors furthermore identified the need for further research on the topic in general and studies of subpopulations using internet only, exclusively land-based and mixed gamblers, in addition to heavy gamblers, gamblers with substance use problems, and ethnicity.

Method gaps were also identified, predominantly regarding the need for longitudinal research, use of improved outcomes measures, and the need for more representative samples. Public health/practical knowledge gaps were under-represented.

### 3.2. Expansion of the Sports Betting Market into the Online Environment

Sports betting, which has become easily accessible due to expansion of the sports betting market into the online environment, and in particular through smartphone applications, is another emerging area raised in the gambling literature, and was the focus in 22 articles. There were three types of studies in this area, namely studies exploring demographic and behavioural characteristics of sports bettors, studies focusing on the fantasy sports phenomenon and those exploring sports betting advertising.

Looking at demographics, an Australian study found being young, male, single, educated, and in full-time employment/study to be risk factors for problem gambling among sports bettors [43]. Additional risk factors were greater frequency and expenditure, greater diversity of gambling involvement and greater impulsivity of responses. An online survey study in Germany identified the typical sports-bettor as male, aged 32, with a low household income, a high interest in sports and willing to take risks [44]. In the Chinese context, Li et al. [45] identified five clusters of sports gamblers (casual players, escalated players, at-risk players, compulsive players and problem players), each with different demographic characteristics.

Fantasy sports is a newly emerging trend in sports betting in the online context, particularly in Canada and the United States where participation in Fantasy Sports leagues is reportedly gaining momentum. Fantasy sports are “an online ancillary to traditional sports media, which allow fans to assume the role of team manager/owner by assembling a virtual line-up of real-world athletes from professional and non-professional sports competitions” [46] (p. 728). Studies into fantasy sports have a range of foci, including characteristics of participants [47], prevalence [48], gambling behaviour [46,49] and problem gambling [49,50]. Gambling participation and problem gambling studies report increased participation in gambling and problem gambling for male fantasy sports players. For example, study of sports-relevant gambling activity in the USA found regular involvement in sports betting, fantasy sports betting and daily fantasy sports betting among adolescents was associated with a higher risk of gambling problems [51].

The influence of sports betting advertising was also identified, particularly in the literature from Australia and the UK. There is growing concern in the literature regarding the effect of sports betting advertising, with 11 studies focused on this aspect. In a series of studies by Pitt and colleagues, the authors discuss sports betting advertisements and other socialisation factors affecting children’s gambling attitudes, behaviours and consumption intentions [52,53,54]. Deans et al. [55] analysed the content of Australian advertisements and identified similarities between sports betting advertising and advertising for other unhealthy commodities. Lopez-Gonzales et al. [56] identified an alignment of alcohol drinking and sports culture in British and Spanish soccer betting adverts. Other studies reported on the normalisation of gambling through sports-related gambling advertising [57,58,59,60] and misleading content [61,62].

#### Gap Analysis

Table 5 shows the distribution of gaps identified in relation to sports betting. Knowledge gaps were again most frequently identified. Authors identified the need for more research on sports betting in general. Further knowledge gaps related to psychological factors, gambling promotion, gambling intentions/behaviours/actions and research focusing on sports betting subpopulations such as women, younger populations, and those with distinct problem gambling behaviour.

In terms of methods gaps, key areas were more representative samples and the need for longitudinal research. Only one article identified a public health/practical knowledge gap relating to research on gambling regulation.

### 3.3. Advertising and Inducements

Linked to the issue of sports betting advertising is new technologies and trends in advertising and inducements, including wagering inducements, loyalty programs, and social media advertising (*n* = 11 articles identified). Two studies on wagering inducements were conducted in Australia. Wagering inducements “incentivise both race and sports betting through offers including sign-up and referral bonuses, ‘free’ bets, matching deposits, cash refunds, bonus odds, happy hours and mobile betting bonuses” [63] (p.686). One study documented the features of inducements [63] and the other involved a survey with 1813 sports bettors to explore whether the uptake of wagering inducements predicted impulse betting, finding this was the case among problem gamblers and frequent sports viewers [64].

There were a further five studies on social media advertising by gambling operators. Gainsbury et al. [65] interviewed 19 individuals working in the gambling industry to explore their use of social media marketing. Social media marketing aimed to attract new customers, strengthen relationships with existing customers and increase customer engagement. Social media marketing included a balance of gambling and non-gambling content with few operators providing specific responsible gambling messages. Gainsbury, Delfabbro, et al. [66] explored the latent messages in social media advertising, finding positive framing and a tendency to encourage gambling. Gainsbury, King, et al. [67] studied the recall and reported impact on gamblers of social media advertising, with data collected through a survey of 964 gamblers who use social media. Results suggest greater impact of social media promotions on problem and moderate-risk gamblers. Concerns about the effect of social media advertising on attitudes and intentions are also seen the work of O’Loughlin and Blaszczynski [68] and Abarbanel et al. [69].

Research on loyalty programs in casinos (*n* = 4) were all marketing-based studies looking at the effect on customer loyalty, reporting varying results of different loyalty program formats [70,71,72,73]. For example, Baluglu et al. [70] found loyalty programs influence transactional outcomes in the casino context.

#### Gap Analysis

Gaps identified in relation to advertising and inducements highlight the relative dearth of research in this area of new technologies and trends (see Table 6). As with other areas identified in this study, knowledge gaps were more represented particularly the need for more research in general and research on gambling outcomes.

In terms of method gaps, these related to qualitative and experimental research, and the need for improved outcomes measures and more representative samples. One article identified a public health/practical knowledge gap relating to the need for research on gambling regulation in relation to advertising and inducements.

### 3.4. Video Gaming and Gambling

A further area of focus in the literature, with 24 articles identified, relates to whether or not there is a relationship between gaming and gambling due to the similarities between the two activities (particularly video gaming and electronic gambling) “due to their similar automated function, immersive experience, and prominent audiovisual elements” [74] (p. 1). This has led to concern about whether video games are associated with an increased likelihood of gambling and problem gambling.

There were four studies exploring this relationship. Results were varied. For example, Molde et al. [75] identified video gaming as a potential gateway behaviour to problem gambling. A study by McBride et al. [76] found adolescents who played video games were significantly more likely to have gambled online for money. Forrest et al. [77] found that the link between video gaming and gambling frequency was related to age, with those who did gamble being slightly older on average than those who did not. Based on a survey of 3942 people who regularly gambled or played videos, Sanders et al. [74] found similar risk factors and manifestations of problem gaming and problem gambling but did not find that involvement in one activity predicted the other. Macey and Hamari [78] explored the relationship between video gaming, spectating e-sports and gambling. The study did not support the notion that video games are in themselves associated with increased potential for problem gambling.

A further issue relates to the convergence of gaming and gambling in terms of the ways in which games are played (particularly player-versus-player in networked sessions), the introduction of gambling-like mechanics through social network games and the expansion of virtual economies and goods [78]. In a commentary article, Drummond and Sauer [79] explored whether video game loot boxes (purchasable randomised rewards) constitute a form of gambling. They sourced a list of games that contain loot boxes available over the past two years and determined the characteristics of the loot boxes. The authors conclude: “in the way they encourage and sustain user engagement, loot-box systems share important structural and psychological similarities with gambling” (p. 532). This raises concerns about the potential for people to migrate from simulated gambling to gambling for money.

Research generally supports the concept of migration to monetary gambling, particularly in young people [80,81,82,83,84,85,86,87,88]. Micro-transactions within simulated gambling were identified as a potential predictor of migration [88]. The two studies addressing this issue had contradictory findings. One study identified micro-transactions as a unique predictor of migration [83], while Hayer et al. [89] did not find micro-transactions to be a significant predictor.

Some studies explored demographic and behavioural factors associated with participation in simulated gambling [90]. Other research notes variation in migration depending on a number of factors. For example, Holingshead et al. [91] explored variation based on motivation. Dussault et al. [92] found migration only holds for poker playing. Hayer et al. [89] found migration for those participating in simulated gambling on social networks (from home) and furthermore noted the role of exposure to advertising for simulated/real games of chance in the decision to gamble for money. Abarbanel et al. [90] also highlighted the potential role of advertising on these platforms.

Armstrong et al. [80], in their literature review, state that current research on the relationship between simulated gambling and gambling for money has tended to be correlational, with further research needed on causal pathways. King and Delfabbro [93] have proposed a two-pathway model conceptualising potential risks and benefits of early exposure to digital simulated activities such as social casino games, although further research is needed. Wohl et al. [88], in their literature review, build on the concept of potential benefits in their review by exploring both the negative aspects of social casino gaming, such as migration to gambling and increased rates of problem gambling, while noting the potential use of social casino gaming as a harm reduction strategy.

#### Gap Analysis

As Table 7 shows, authors saw the need for more knowledge about the relationship between video gaming and gambling. They identified the need for more research on the topic in general as well as research on specific factors, particularly intentions/behaviours/actions, psychological factors and gambling outcomes.

Method gaps related in particular to the need for improved outcome measures, qualitative and longitudinal designs, and more representative samples. Few public health/practical knowledge gaps were identified.

### 3.5. Electronic Gaming Machines

Electronic gaming machines (EGMs) refers to land-based gaming machines, as distinct from online EGMs which are usually called online slots. They are the outcome of digital technological advancements in the field of audio-visual effects. These machines have raised particular concerns due to their availability worldwide and their association with problem gambling [94]. A total of 15 articles addressed various aspects of EGMs.

Armstrong et al. [95] conducted an environmental scan to explore features that might encourage reckless betting or entice new players. They conclude that technological enhancements have the potential to “increase immersion and potentially encourage elevated play by automatic betting functions, reducing the time between games and reinforcing betting behaviours with intricate graphics, animations and sound” (p. 120). Goodwin et al. [96] conducted a qualitative study of player preferences in relation to traditional and innovated gambling products. Traditional products were seen as less harmful, more social, and more enjoyable than innovated products. In a critical review, Armstrong et al. [97] state that more research is needed in this area, particularly in terms of the effect on player behaviour and potential risks.

Looking at particular characteristics of EGMs, the research on jackpots identified that various characteristics increase gambling behaviour and affect gambling motivation. Browne et al. [98] found that jackpots influenced gambling motivations, with this being particularly the case for gamblers at-risk of problem gambling. They also found an association of jackpots with greater spend. Similarly, Li et al. [99] found intensifying effects on gambling behaviour with high value jackpot configurations. Donaldson et al. [100] note a marginal positive contribution of hidden jackpots to risky playing behaviour.

Considering the potential effects across a range of EGM characteristics, Landon et al. [101] conducted a focus group study in New Zealand exploring which characteristics participants found attractive. Free spin features were the most attractive. Greater intensity and duration of gambling were found to be associated with smaller win related characteristics and low-denomination machines with multiple playable lines. The effect of multi-line play on attention was also found in a study by Murch and Clark [102].

Barton et al. [94] conducted a systematic review of losses disguised as wins and near misses in EGMs. They found near misses motivated continued play but had varied effects on betting behaviour and players’ emotional states, while losses disguised as wins were related to an overestimation of how much a player is winning. The study on speed of play looked at the potential value of using a measure of individual rates-of-play in EGM research [103].

An article on gambling machine annexes explored the characteristics of annexes and discussed how they might promote heavy and problematic gambling [104]. Similarly, an article on anthropomorphisation of slot machines explored its possible negative impact [105], and a discussion article on virtual reality [106] noted that there is little research on this form of gambling and explored public policy implications.

#### Gap Analysis

The gap analysis, presented in Table 8, followed a similar trend to gaps identified in other areas of gambling technologies and trends. Knowledge gaps were most frequently identified, particularly the need for research exploring psychological factors and more research in general. Few method gaps were identified in terms of using data mining and including more representative samples. In terms of public health/practical knowledge gaps, one article noted the need for research into gambling regulation related to EGM technologies and trends.

## 4. Discussion

This gap analysis sought to answer the question: What are the gaps in our understanding of emerging technologies and new trends in gambling? Understanding the emerging landscape of gambling is important in informing a public health approach to addressing the harms associated with gambling. We identified 116 peer-reviewed articles published since 2015. Key emerging technologies and new trends include Internet gambling, followed by those exploring the relationship between gaming and gambling, the expansion of the sports betting market, emerging trends in the development of Electronic Gambling Machines (EGMs) and new technologies and trends in advertising and inducements.

Across the body of work, the main gap areas related to gaps in knowledge about emerging digital technologies and new trends in gambling. There was a strong call for more research in this area in general, in addition to the need for research on subpopulations such as those gambling using different formats, those with varying levels of problem gambling, and vulnerable populations such as young people and minority ethnic groups. From a methods perspective, researchers saw the need for longitudinal studies, qualitative research, and improved outcome measures. There was less focus on public health and practical knowledge gaps relating to responsible gambling initiatives and policy/regulation focused on new technologies and trends.

Internet (or online) gambling was the most commonly researched emerging technology/trend. While greater rates of problem gambling were associated with this form of gambling, recent research suggests gambling via the Internet is not inherently problematic but rather appears to affect different gamblers in different ways. In analysing gaps identified in the literature, research indicating a heterogeneity of Internet gamblers and differences in risk profiles is an important finding as it suggests key areas of focus to best support those at risk of problem gambling, rather than assuming a gambling modality is intrinsically problematic for all gamblers. The research is predominantly cross-sectional survey research. More focused research is needed comparing online versus land-based gamblers, and problem versus non-problem Internet gamblers, according to different characteristics (e.g., level of problem gambling/risk; type of gambling; populations; attitudes/behaviours). Further research is also needed on which features of Internet gambling contribute to gambling disorders and gambling-related harm. The potential for greater problems associated with gambling via mobile/smartphone was identified and this is an important area for future research. The ‘hidden’ nature of risk and harm, and challenges in monitoring online gambling were particularly noted and are therefore important areas for future research. With greater online gambling opportunities, more knowledge is needed about the impact of impulsive opportunistic gambling.

In terms of simulated gambling, research generally supports the concept of migration to monetary gambling, particularly by young people. Longitudinal studies are needed to further explore their migration to gambling for money. The emergence of Cryptocurrency and Block Chain will likely also warrant research. The link between video gaming in general and gambling (in terms of whether engagement/problems in one activity is associated with engagement/problems in the other) is still not clear, suggesting the need for further research. Gaps in this area relate to research methods and focus. Longitudinal studies are needed on the migration to gambling for money. Hayer et al. [89] (p. 943) note, “future studies must consider the complex notion of the gateway effect including (simulated/real) gambling type, mode of access and a more precise definition of migration”. The link between engagement in/problems with video gaming and engagement in/problems with gambling is still not clear, suggesting the need for further research.

Changes to the structure and operation of electronic gambling machines (EGMs) is another emerging technology/trend identified in the literature; in particular, in relation to various features of EGMs that may increase or decrease harm to gamblers. Limited research attention was evident in this area of research since 2015, and researchers tended to focus on individual features. A more systematic approach covering the effect of various features, both individually and in combination might assist in the understanding of which specific venue feature contributes to gambling disorders and harm. The research included here identifies potential harms associated with automation of traditional games, EGM jackpots, multi-line play, speed of play, annexes, and anthropomorphising games. However, the research covers a range of characteristics and gamblers. A gap in the literature is research to fully understand which features contribute to gambling disorders and harm.

Research evidence on the effects of digitally enhanced EGM features are limited. Specifically, there is a gap in the literature examining the liability of EGM designers who conceal the information related to potentially harmful and addictive nature of EGMs (similar to the tobacco industry concealment of the dangers of smoking).

Another significant and emerging area of future research relates to sports betting and the effects of the rapidly expanding sports betting market and its effect on gambling prevalence and problem gambling. Another related area of future research is the effect of advertising on gambling attitudes, behaviours, and gambling-related harm. The gap analysis here notes the current focus is on profiling those who engage in sports betting and fantasy sports and identifying harms association with these activities, along with the growing body of research analysing sports betting advertising. More research is needed on understanding and mitigating harms associated with sports betting and in particular sports betting advertising.

Overall, research highlights potential risks associated with the normalisation of gambling through advertising and the potential impact of advertising and inducements on gambling behaviour, particularly for problem/at-risk gamblers. However, the association between advertising and inducements and rates of gambling and problem gambling have not been established using more robust methods. Further research on new technologies and trends in advertising and inducements from other countries would inform the public health and policy debate in this area, particularly given different regulatory environments relating to advertising. Furthermore, there is no research examining the effect that advertising control has on harms secondary to gambling. Finally, there is growing interest in the role of loyalty programs in terms of both their potential for harm and possibilities for harm minimisation. There is limited research on the role of loyalty programs, identifying the need for empirical research on antecedents and consequences of these programs, in addition to exploration of their potential in terms of harm minimisation.

We identify three further areas where research is needed across the body of research into emerging digital technologies and new trends. While not a significant focus in gaps identified in the literature, we suggest the need for research focusing on responsible gambling initiatives. There is also a need to shift the focus from the level of the individual gambler to exploring what a public health approach to addressing the issues associated with new gambling technologies and trends.

### Limitations

Some limitations were placed on the methodology used to search the literature including the omission of literature published prior to 2015, not searching reference lists and exclusion of grey literature. A more general limitation likely arises from considering gaps from the perspective of what already exists. This may have introduced an inherent bias by relying on the gaps and recommendations for further research that particular experts and research teams promoted as important. However, consultation between subject matter experts on our research team and the ORG to identify further gaps not identified in the body of research likely mitigates this limitation to some extent. Despite the wide search terms used, some peer-reviewed articles that could have met the inclusion criteria may not have been captured. We also acknowledge that different gambling cultures, different regulatory frameworks, and variations in prevalence of online gambling may have affected not only the researchers’ research priorities but also the gaps identified. This is important to take into account when interpreting the published articles and the gaps identified. Variations according to age and gender were not the focus of this gap analysis, and would require further dedicated review.

## 5. Conclusions

Emerging technologies and trends in gambling have the potential to significantly affect rates of gambling and problem gambling. This gap analysis presents the results of a systematic approach to reviewing current trends and identifying gaps, with a view to inform the future research agenda in this area. It is clear that further research is needed on the implications for gamblers and the wider community of emerging technologies and new trends in gambling in the digital age. The current gap analysis indicates that this should include more research on the potential for online gambling to affect different subpopulations of gamblers in different ways, the hidden nature of risk and harm and challenges in monitoring online gambling, and more longitudinal studies of the impact of gaming, specifically on the migration to gambling for money. The gap analysis also indicates that further research is needed on the potential harm arising from technological developments associated with EGMs, from online sports betting, and the impact of advertising, inducements and loyalty programs which have a pervasive presence as part of emerging technologies associated with gambling. The development and testing of a public health approach to addressing the harms associated with gambling in these areas is needed.

## Figures and Tables

**Figure 1 ijerph-17-00744-f001:**
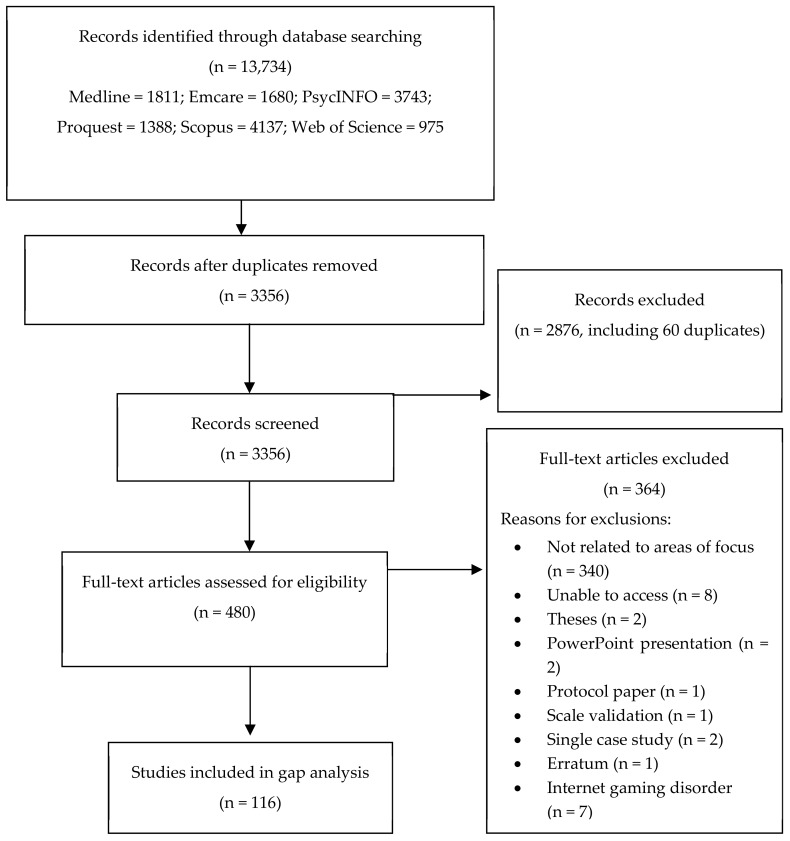
PRISMA diagram.

**Table 1 ijerph-17-00744-t001:** Inclusion and exclusion criteria.

Inclusion Criteria	Exclusion Criteria
English language	Papers not in English
Peer-reviewed journal articles	Books, conference presentations, PhD theses/dissertations, PowerPoint presentations and posters, government reports
The paper reports on an empirical study or systematic review/commentary/discussion article addressing the topic	Protocols, single case studies
Published 1 January 2015–2016 October 2018	Published prior to 2015
The focus of the paper is on gambling and the paper addresses emerging technologies or new trends	The focus of the paper is not on gambling The paper addresses new technologies and trends in treatment for gambling problems

**Table 2 ijerph-17-00744-t002:** Typology of gaps.

Gap Type	Areas for Further Research
Method gaps	Longitudinal research
	Experimental research
	Qualitative research
	Use of improved outcome measures
	Integration between studies
	Using of data mining
	More representative samples
Knowledge gaps	More research in general (e.g., updating data, deeper analysis of associations)
	Research on specific factors (e.g., psychological factors; gambling intentions/behaviours/actions; gambling outcomes; gambling promotion; social factors)
	Studies of subpopulations
	Replication in other locations
	Research on other/related technologies/trends
	Applicability to other gambling types
	Applicability to other disorders
	Applicability to other populations
Public health/Practical knowledge gaps	Responsible gambling policies/initiatives (implications for/effectiveness of)
	Gambling regulation (effectiveness of/relationship with)

**Table 3 ijerph-17-00744-t003:** Country of origin for included articles.

No.	Country of Origin	No. Research Articles	No. Discussion/Review/Other Articles	Percentage of Overall Included Articles
1	Australia	39	7	39
2	Canada	15	2	14
3	United States of America (USA)	14		12
4	United Kingdom (UK)	4	1	4
5	China	4		3
6	France	4		3
7	Germany	4		3
8	Greece	1	1	2
9	Italy	3		3
10	New Zealand	1	1	2
11	Norway	2		2
12	Spain	2		2
13	Switzerland	1		1
14	UK and Spain	2		2
15	Austria	1		1
16	Denmark	2		2
17	Finland	1		1
18	Iceland	1		1
19	Portugal		1	1
20	Singapore	1		1
21	Taiwan	1		1
	**Totals**	**103**	**13**	**100**

**Table 4 ijerph-17-00744-t004:** Internet gambling gaps.

Gap Type	No. of Times Identified in the Articles
**Method gaps**	**19**
Longitudinal research	6
Use of improved outcome measures	4
More representative samples	4
**Knowledge gaps**	**42**
More research in general	9
Research on specific factors—psychological factors	12
Research on specific factors—gambling intentions/behaviours/actions	7
Research on specific factors—gambling outcomes	3
Studies of subpopulations	7
**Public health/Practical knowledge gaps**	**5**
Responsible gambling policies/initiatives	2
Gambling regulation	3

**Table 5 ijerph-17-00744-t005:** Sports betting gaps.

Gap Type	No. of Times Identified in the Articles
**Method gaps**	**4**
Longitudinal research	2
More representative samples	2
**Knowledge gaps**	**31**
More research in general	5
Research on specific factors—psychological factors	7
Research on specific factors—gambling intentions/behaviours/actions	3
Research on specific factors—gambling outcomes	1
Research on specific factors—gambling promotion	4
Research on specific factors—social factors	2
Studies of subpopulations	3
Replication in other locations	2
Research on other/related technologies/trends	1
Applicability to other gambling types	2
Applicability to other populations	1
**Public health/Practical knowledge gaps**	**1**
Gambling regulation	1

**Table 6 ijerph-17-00744-t006:** Gambling advertising and inducements gaps.

Gap Type	No. of Times Identified in the Articles
**Method gaps**	**4**
Experimental research	1
Qualitative research	1
Use of improved outcome measures	1
More representative samples	1
**Knowledge gaps**	**10**
More research in general	3
Research on specific factors—psychological factors	1
Research on specific factors—gambling intentions/behaviours/actions	1
Research on specific factors—gambling outcomes	2
Research on specific factors—gambling promotion	1
Studies of subpopulations	1
Replication in other locations	1
**Public health/Practical knowledge gaps**	**1**
Gambling regulation	1

**Table 7 ijerph-17-00744-t007:** Video gaming and gambling gaps.

Gap Type	No. of Times Identified in the Articles
**Method gaps**	**14**
Longitudinal research	2
Experimental research	1
Qualitative research	3
Use of improved outcome measures	5
Using of data mining	1
More representative samples	2
**Knowledge gaps**	**24**
More research in general	10
Research on specific factors—psychological factors	3
Research on specific factors—gambling intentions/behaviours/actions	3
Research on specific factors—gambling outcomes	3
Research on specific factors—gambling promotion	1
Studies of subpopulations	4
**Public health/Practical knowledge gaps**	**2**
Responsible gambling policies/initiatives	1
Gambling regulation	1

**Table 8 ijerph-17-00744-t008:** Electronic gaming machines gaps.

Gap Type	No. of Times Identified in the Articles
**Method gaps**	**2**
Using of data mining	1
More representative samples	1
**Knowledge gaps**	**12**
More research in general	4
Research on specific factors—psychological factors	5
Research on specific factors—gambling outcomes	1
Research on specific factors—gambling intentions/behaviours/actions	1
Studies of subpopulations	1
**Public health/Practical knowledge gaps**	**1**
Gambling regulation	1

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
