# Peer review of "A Literature Review and Gap Analysis of Emerging Technologies and New Trends in Gambling"

_ijerph, 2020, doi:10.3390/ijerph17030744_

Round 1

Reviewer 1 Report

The paper in its simplicity is original and interesting, the authors have analyzed over a hundred papers to extract information of interest for research.
I would ask the authors to describe the regions for which 2876 papers, including 60 duplicates have been excluded, it is not clear whether they did not meet the inclusion criteria and which ones.

The gap analysis was carried out with rigor and clarity, however the motivation for the authors to analyze some items instead of others is unclear, I would ask them to explain the reasons that led to the choice. The discussion is very interesting and clear.

Author Response

Reviewer 1

The paper in its simplicity is original and interesting, the authors have analyzed over a hundred papers to extract information of interest for research.
I would ask the authors to describe the regions for which 2876 papers, including 60 duplicates have been excluded, it is not clear whether they did not meet the inclusion criteria and which ones.

Response: We thank the reviewer for their supportive comments. It is not standard practice to provide this level of description of reasons for exclusion at this first stage of the review process (screening by title), as per PRISMA guidelines. Also, duplicates were excluded for that reason; because they were duplicates. The 2876 papers were excluded strictly for the various reasons stated in Table 1. They were from various countries, including Australia, the USA and Canada and the many other countries from which included studies also originated. We have provided a detailed list of reasons for exclusions at the full-text screening stage within Figure 1.

The gap analysis was carried out with rigor and clarity, however the motivation for the authors to analyze some items instead of others is unclear, I would ask them to explain the reasons that led to the choice. The discussion is very interesting and clear.

Response: Our motivation to analyse the items listed across 3.1 – 3.5 was determined by the results of the literature search itself. Following use of the broad and inclusive search terms (see lines 85-88) to search the 6 electronic databases, then the systematic screening processes undertaken by the research team to arrive at the 116 included papers, the 5 items/areas capturing various aspects of new technologies and trends became evident as part of grouping these 116 papers (see lines 108-110). We have revised this section slightly to make it clear that the 5 areas of focus outlined in the results arose from the review process and then grouping in these main areas of apparent focus.

Reviewer 2 Report

I found this manuscript as a very interesting contribution to the literature. It is well-written and well-organized. It is also valuable because it informs the future research agenda in the area.

I only have a few minor remarks worth consideration.

Research articles’ country of origin is important to know when evaluate the gaps stated. The authors mention that 32 % of the articles were published in Australia. It is obvious, that different gambling cultures, different regulatory frameworks, prevalence of mobile gambling etc. affect not only to the researchers’ research priorities but also to the gaps identified. This is important to take into account when interpreting the published papers and gaps identified. Due to the international audience of the journal, I strongly suggest that the authors reflect on this issue in the Discussion.

Add clear information (i.e. time frame, month/year) into the Material and Methods section when the literature search was performed.

row 90; “three reviewers conducted the initial screening…”. Please add more specific information whether the reviewers were researchers/senior investigators or trained members of the research team or who they were.

the minor note – it is not always clear within the text whether the authors talk about physical EGMs and/or online gambling on EGMs. Please specify where needed.

There are some issues worth mentioning within the Discussion limitation section, but which have not been mentioned there. For example, the quality of the papers included for the study was not performed and it appears (at least for me) that despite wide search terms used some peer-reviewed articles that would have met the inclusion criteria were not captured.

Good luck with the promising manuscript!

Author Response

Reviewer 2

I found this manuscript as a very interesting contribution to the literature. It is well-written and well-organized. It is also valuable because it informs the future research agenda in the area. I only have a few minor remarks worth consideration.

Research articles’ country of origin is important to know when evaluate the gaps stated. The authors mention that 32% of the articles were published in Australia. It is obvious, that different gambling cultures, different regulatory frameworks, prevalence of mobile gambling etc. affect not only to the researchers’ research priorities but also to the gaps identified. This is important to take into account when interpreting the published papers and gaps identified. Due to the international audience of the journal, I strongly suggest that the authors reflect on this issue in the Discussion.

Response: We have revised the manuscript, providing more detail of the breakdown of countries from which the articles were published. Given that the 116 papers come from 21 countries, and for readability, we have provided specific detail about the % for the 3 countries with the greatest % only, and added a table summarising the complete breakdown of countries.  Most articles were from Australia (32%), and the next largest contribution came from Canada (15%), then USA (14%); the remaining 18 countries contributed less than 10% each. As per the reviewer’s suggestion, we have also added further comment in the discussion limitations section as follows: “We also acknowledge that different gambling cultures, different regulatory frameworks, and variations in prevalence of online gambling may have affected not only the researchers’ research priorities but also the gaps identified. This is important to take into account when interpreting the published articles and the gaps identified.”

Add clear information (i.e. time frame, month/year) into the Material and Methods section when the literature search was performed.

Response: The actual search of databases was conducted on 16th October 2018. The team conducted the review between October 2018 and March 2019. We have added this information to the manuscript. See lines 88-89 within the Materials and Methods section.

row 90; “three reviewers conducted the initial screening…”. Please add more specific information whether the reviewers were researchers/senior investigators or trained members of the research team or who they were.

Response: The reviewers were the first three authors of the manuscript; this included two senior researchers with previous experience in conducting several literature reviews, and a senior researcher with specific expertise in problem gambling treatment and research, and with some prior experience in conducting literature reviews.

the minor note – it is not always clear within the text whether the authors talk about physical EGMs and/or online gambling on EGMs. Please specify where needed.

Response: EGM refers only to land-based electronic gaming machines. Usually online EGMs are called online slots. We have clarified that electronic gaming machines (EGMs) refers to land-based gaming machines. See lines 347-348.

There are some issues worth mentioning within the Discussion limitation section, but which have not been mentioned there. For example, the quality of the papers included for the study was not performed and it appears (at least for me) that despite wide search terms used some peer-reviewed articles that would have met the inclusion criteria were not captured.

Response: We did perform quality ratings for all included peer-reviewed studies. We have now provided summary information about these in the manuscript (see lines 109-115). The quality rating tables are included in supplementary materials. The process found that all peer-reviewed research was generally well conducted and met the rating criteria. No studies were excluded due to poor quality. We have also included a general statement in the limitations: “Despite the wide search terms used, some peer-reviewed articles that could have met the inclusion criteria may not have been captured.”

Reviewer 3 Report

The aim of the manuscript was to identify the gaps in emerging technologies and new trends related to gambling. The gap analysis of 116 articles from 2015 on Internet gambling, he relationship between video gaming and gambling, sports betting market, Electronic Gambling Machines, and new technologies and trends in advertising and inducements, suggests the emergence of gaps related to the need for more research in general, on specific gambling sub-populations, and vulnerable populations, and also the need for longitudinal studies, qualitative research, improved outcome measures, and the development and testing of a public health approach to addressing the harms associated with gambling in these areas.

I find the following critical points:

(Note that the points are in the order of the manuscript, not in the order of their importance).

In my view, in the Introduction, Authors have not to indicate that this study has been committed by the Office of Responsible Gambling (ORG) of the New South Wales (NSW), otherwise this work may seem a research report for a specific funding, more than material eligible for a research article. Consequently, Authors have to better justify the reasons why there is a need, in terms of research and practice, for the reported research question, i.e., what are the gaps in our understanding of emerging technologies and new trends in gambling? They also must better explain why they chose Internet gambling, he relationship between video gaming and gambling, sports betting market, Electronic Gambling Machines, and new technologies and trends in advertising and inducements, as specific main themes of the article proposed. To that aim, the Introduction may benefit from a wide description of how the new emerging technologies are linked to gambling opportunities nowadays and also a description of gambling prevalence of online gambling for different types of gamblers, based on gender and age. I am not very friendly with the gap analysis but Authors have to better justify why they decided to do a gap analyses rather than a systematic review. Then, Authors must report some conventional Guidelines to conduct the gap analysis (as the PRISMA guidelines for the systematic review), otherwise the Authors’ report may be more similar to a synthesis of indications for future studies reported at the end of each article considered in this work. In this regard, in my view, the article considered must be reported in the paper more than in the supplementary material. Authors may organize them into some thematic tables. The Introduction must be more focused on the advantages to conduct this gap analysis. More justifications on why 2015 has been chosen as timeframe must be added. Are systematic reviews included in the gap analyses? From Table 1 it does not seem, but then one systematic review is cited. I have some difficulties in understanding Table 2 reporting the typologies of gaps. Isn’t that a result of the gap analysis itself? Authors reported that for 8 articles they had not have access to the text. Authors must specify if they had written to the authors. I have some criticism for the Results section. The results are presented in a confusing way. In my view, it would be better to first describe the results by the main theme. In this regard more organization should be placed in reported the results also by age (children, adolescents, adults). No studies are reported concerning older adults. Are there any findings about their involvement in new forms of gambling? If not, this aspect may be included in the gap analysis. In my view, after having reported all these results, the overall gaps, considering the main themes, should be highlighted. Conclusions must be reinforced, for example by more in detail specify research lines to overcome the recognized gaps, and also by driving practical indications concerning what research already knows.

Author Response

Reviewer 3

The aim of the manuscript was to identify the gaps in emerging technologies and new trends related to gambling. The gap analysis of 116 articles from 2015 on Internet gambling, he relationship between video gaming and gambling, sports betting market, Electronic Gambling Machines, and new technologies and trends in advertising and inducements, suggests the emergence of gaps related to the need for more research in general, on specific gambling sub-populations, and vulnerable populations, and also the need for longitudinal studies, qualitative research, improved outcome measures, and the development and testing of a public health approach to addressing the harms associated with gambling in these areas.

I find the following critical points: 

In my view, in the Introduction, Authors have not to indicate that this study has been committed by the Office of Responsible Gambling (ORG) of the New South Wales (NSW), otherwise this work may seem a research report for a specific funding, more than material eligible for a research article. Consequently, Authors have to better justify the reasons why there is a need, in terms of research and practice, for the reported research question, i.e., what are the gaps in our understanding of emerging technologies and new trends in gambling? They also must better explain why they chose Internet gambling, he relationship between video gaming and gambling, sports betting market, Electronic Gambling Machines, and new technologies and trends in advertising and inducements, as specific main themes of the article proposed. To that aim, the Introduction may benefit from a wide description of how the new emerging technologies are linked to gambling opportunities nowadays and also a description of gambling prevalence of online gambling for different types of gamblers, based on gender and age.

Response: We thank the reviewer for their critical comments. We believe that the focus on online gambling for this article is worthwhile, given the prolific growth of online gambling and growing concern for its potential harms (For an example where this focus is argued as important and timely, see: McCormack A, Shorter GW, Griffiths MD. An empirical study of gender differences in online gambling. J Gambl Stud 2014; 30:71–88). Growth of online gambling has also partly contributed to the growth of sports betting (please see: AGRC Discussion Paper No. 4 – November 2014). The focus on online gambling arose because it is a rapidly evolving area of gambling that has brought new challenges globally for regulators and policy makers, in addition to communities and problem gambling treatment providers, requiring potentially new approaches to address gambling harm from online gambling. We did not choose internet gambling, instead we conducted a broad review of the current research on gambling to identify emerging trends and new technologies. These identified technologies and trends included Internet gambling but also technologies such as virtual reality and other technologies and trends identified in the review (technological changes to EGMs, sports betting, advertising, social casino games). We tried to make clear in the Intro that Internet gambling is only one of many emerging technologies and new trends.

We have expanded the introduction to include more information about why we felt this focus on online gambling was important to review (see lines 44-48). We are conscious that the paper is already lengthy and so have made a general statement here. Prevalence of various forms of online gambling was not the focus of this component of the review and the existing research literature offers only very limited conclusions about this, given the difficulty and challenges in accurate measurement.

I am not very friendly with the gap analysis but Authors have to better justify why they decided to do a gap analyses rather than a systematic review. Then, Authors must report some conventional Guidelines to conduct the gap analysis (as the PRISMA guidelines for the systematic review), otherwise the Authors’ report may be more similar to a synthesis of indications for future studies reported at the end of each article considered in this work. In this regard, in my view, the article considered must be reported in the paper more than in the supplementary material. Authors may organize them into some thematic tables. The Introduction must be more focused on the advantages to conduct this gap analysis.

More justifications on why 2015 has been chosen as timeframe must be added.

Response: As stated in the manuscript, 2015 was the date set by the funder of this research. We believe that we have provided detail about the guidelines used to conduct the gap analysis at the beginning of the Materials and Methods section where we outline our approach involving 6 processes modified from Otto et al. We have organised the gaps into 5 areas, based on the findings of the review. (akin to what the reviewer refers to as ‘themes’. Please also see our response to reviewer 1 where we clarify that section 3.1 – 3.5 (the 5 gap areas) were determined by the results of the literature search itself. Following use of the broad and inclusive search terms (see lines 81-84) to search the 6 electronic databases, then the systematic screening processes undertaken by the research team to arrive at the 116 included papers, the 5 items/areas capturing various aspects of new technologies and trends became evident as part of grouping these 116 papers (see lines 106-110).

Are systematic reviews included in the gap analyses? From Table 1 it does not seem, but then one systematic review is cited.

Response: We have revised Table 1 to clarify that systematic reviews were included.

I have some difficulties in understanding Table 2 reporting the typologies of gaps. Isn’t that a result of the gap analysis itself?

Response: Table 2 provides a breakdown of the overall typology of gaps. This typology was derived from the larger review. We then used this typology as a basis for reporting within each specific gambling type/area of focus. We have revised the text to clarify this point.

Authors reported that for 8 articles they had not have access to the text. Authors must specify if they had written to the authors.

Response: As this was a rapid review, we did not write to the authors. We have provided a statement about this in the manuscript (see lines 104-105)

I have some criticism for the Results section. The results are presented in a confusing way. In my view, it would be better to first describe the results by the main theme. In this regard more organization should be placed in reported the results also by age (children, adolescents, adults). No studies are reported concerning older adults. Are there any findings about their involvement in new forms of gambling? If not, this aspect may be included in the gap analysis.

Response: The focus of this review was new technologies and trends in gambling, in general; it was not to undertake a more fine-grained analysis of age differences in online gambling. We suggest that that would require its own dedicated review with different research questions. Gender differences is a further area of focus that would similarly be best suited to a specific review. We have added some further comments to the limitations about age and gender (see lines 483-489).

In my view, after having reported all these results, the overall gaps, considering the main themes, should be highlighted. Conclusions must be reinforced, for example by more in detail specify research lines to overcome the recognized gaps, and also by driving practical indications concerning what research already knows. 

Response: We have reviewed the discussion and conclusion sections of the manuscript and added further statements to highlight the main findings of this review. Please see conclusions section (lines 495-502).

We have not provided further detailed specific research lines to overcome the gaps, beyond the information that we have already outlined in the various tables, as this was not the purpose of the paper and we feel that it is not our place to tell researchers how they should conduct such research; its purpose was to identify the gaps that emerged through our synthesis of the 116 articles, including the gaps that were identified by the authors of the articles themselves.

Round 2

Reviewer 3 Report

Authors have substantially follow my indication in revising the paper. However, I suggest to more deeply revise the Introduction by reporting in the paper the argumentations and citations they reported in the response letter concerning online gambling.

Moreover, why do Aithors leave in the text that this study has been committed by the Office of Responsible Gambling (ORG) of the New South Wales (NSW)?

Author Response

Response to Reviewer 3

Authors have substantially followed my indication in revising the paper. However, I suggest to more deeply revise the Introduction by reporting in the paper the argumentations and citations they reported in the response letter concerning online gambling.

Response:

We thank the reviewer for their critical comments. We have added the following text to the introduction, drawing on the ideas with noted in the previous response. Please see lines 44-50. As part of this change, we added these references to the reference list and adjusting the reference numbering throughout the paper.

“This prolific growth in online gambling has been accompanied by growing concern for its potential harms [6]; hence reviewing the current literature and evidence gaps is important and timely. Growth of online gambling has also been implicated in the general growth in gambling, more broadly; for example, in the growth of sports betting among people who did not previously gamble [7]. With such growth and potential greater diversity in gambler populations, there is likely growth in new populations groups experiencing problem gambling.”

Moreover, why do Authors leave in the text that this study has been committed by the Office of Responsible Gambling (ORG) of the New South Wales (NSW)?

Response: This as an important part of the justification for the study and the parameters determined within it. Clarification of the role of the ORG and its relationship to the Responsible Gambling Fund (RGF) was also a condition of the funding of this research and reporting of results in any publications. That is, the ORG has requested that we include the description of the context for transparency purposes. Gambling research funding can be a contentious issue, particularly where research funds may arise via monies from gambling. This concern is a bit like the potential for smoking research that might be funded by the Tobacco industry. We have therefore left this information unchanged in the paper. We hope that you find this explanation sufficient.

We have also added URL for information about the MMAT quality rating scale used in the study, as we noticed that its inclusion was overlooked in the previous revision.